# Synthesizing and Simulating Volumetric Meshes from Vision-based Tactile Imprints

Xinghao Zhu[1,2], Siddarth Jain[2,*], Masayoshi Tomizuka[1], and Jeroen van Baar[2]

*Abstract*—**Vision-based tactile sensors typically employ a deformable elastomer and a camera to provide high-resolution contact images. This work focuses on learning to simulate and synthesize the volumetric mesh of the elastomer based on the image imprints acquired from tactile sensors. Obtaining accurate volumetric meshes for the elastomer can provide direct contact information and benefit robotic grasping and manipulation. Our method [1] proposes a train-then-adapt way to leverage synthetic image-mesh pairs and real-world images from finite element methods (FEM) and physical sensors. Our approach can accurately reconstruct the deformation of the real-world tactile sensor elastomer in various domains. While the proposed learning approaches have shown to produce solutions, we discuss some limitations and challenges for viable real-world applications.**

## I. INTRODUCTION

Tactile sensing is essential for humans when interacting with environments. Robotic tactile sensors can provide contact profiles during grasping and manipulation tasks. Among different designs, vision-based tactile sensors are variants [2]–[9]. They use cameras to capture the deformation of the contact elastomer with high-resolution images, as shown in Fig. 1 (a) and (b).

Representing the deformable elastomer with volumetric mesh can advance the development of vision-based tactile sensors. Volumetric meshes provide accurate and direct information about the contact. This information can benefit manipulation tasks like in-hand object localization [10]–[12], vision-free manipulation [13]–[15], and contact profile reconstruction [4], [5], [16]–[18]. Moreover, meshes can be used for precise dynamics learning [19]–[21] and future state prediction [22], [23].

Our method directly predicts the volumetric mesh from images using vision-based tactile sensors, such as the Gel-Slim [5], in a sim-to-real setting. We first employed supervised training to synthesize the volumetric mesh from image imprints gathered from synthetic 3D FEM. However, directly deploying the network into the real-world yields poor reconstruction results due to the sim-to-real gap. Thus, we propose data augmentations and a self-supervised adaptation method on real-world images to address this gap. Experiments demonstrate that the proposed method can transfer networks for sim-to-real, seen contact objects to novel contact objects, and between different GelSlim sensor instances (as shown in Fig. 1 (c) and (d)).

* Corresponding Author
[1] Mechanical Systems Control Lab, UC Berkeley, Berkeley, CA, USA. {zhuxh,tomizuka}@berkeley.edu
[2] Mitsubishi Electric Research Laboratories (MERL), Cambridge, MA, USA {sjain,jeroen}@merl.com

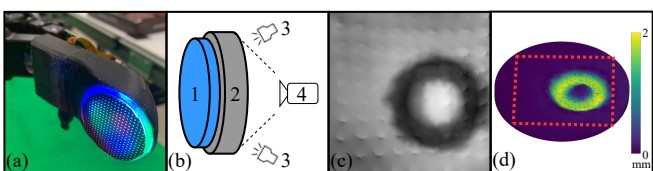

Fig. 1: **(a)** the GelSlim visual-tactile sensor, **(b)** the construction of the sensor, with the elastomer (1), the transparent lens (2), the lights (3), and the camera (4). **(c)** a depth image observation obtained from the sensor, and **(d)** the corresponding reconstructed volumetric mesh with our method. The red rectangle denotes the camera's view range, and the color represents the displacement level.

In the remaining of this paper, we illustrate the method and demonstrate the results in Section II and Section III. Then we discuss the limitations and future directions in Section IV.

## II. METHODS

This section first introduces the problem statement and preliminaries. Next, the image-to-mesh projection and self-supervised adaptation methods are discussed. Finally, the datasets are described, including synthetic labeled data, real-world unlabeled data, and data augmentation techniques.

### A. Problem Statement and Preliminaries

This paper focuses on the problem of reconstructing an elastomer's volumetric mesh with image observations. The non-injective projection from surface images to volumetric vertex makes this problem nontrivial. Some preliminaries are:

*1) Image Observations:* Visual tactile sensors contact objects with a silicone elastomer and use a camera to capture the deformation of the surface, as shown in Fig 1. The captured RGB image can be used to construct a depth map using shape from shading [5], [24]. Compared to raw RGB images, depth maps can better represent the geometry of the contact surface and are much easier to simulate using synthetic cameras. Thus we use $(128 \times 128)$ depth maps $I$ as the image observations in this paper.

*2) Volumetric Meshes with FEM:* In the FEM, geometrical shapes are represented by volumetric meshes $\mathcal{M}$. With high-resolution meshes and small computation steps, FEM can estimate the forward dynamics of soft bodies [19], [25]. This paper uses graphs to represent volumetric meshes. Specifically, volumetric meshes are defined as a set of vertices and edges, $\mathcal{M} = (\mathcal{V}, \mathcal{A})$, with $n$ vertices in 3D Euclidean space, $\mathcal{V} \in \mathbb{R}^{n \times 3}$. The adjacency matrix $\mathcal{A} \in \{0, 1\}^{n \times n}$ represents the edges.

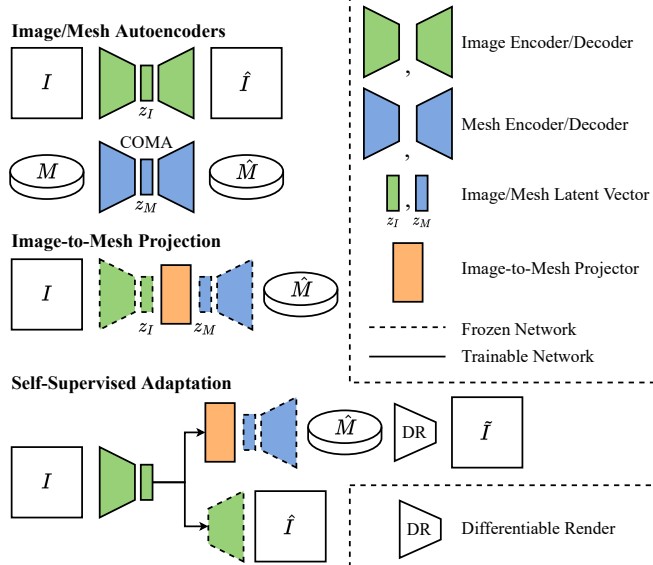

Fig. 2: Training structure. The image-to-mesh projection network is optimized with pre-trained autoencoders. The self-supervised adaptation transfers the projection network to various domains with a differentiable render.

### B. Supervised Image-to-Mesh Projection

The image-to-mesh projection is learned with latent representations. Fig. 2 shows the training structure of the network. The image variational autoencoder (VAE) reconstructs depth maps $I$ to $\hat{I}$ and is trained as a $\beta$-VAE. We adopt the convolutional mesh autoencoders (COMA) [26] for the volumetric mesh VAE. COMA uses spectral graph convolutions [27] to extract features and a hierarchical pooling operation to reduce vertices. The latent projection model is comprised of three fully connected layers. It is trained in a supervised manner with the encoder and decoder frozen.

### C. Self-Supervised Adaptation

When deploying the trained network to the real world, covariate shift problems may reduce the performance significantly [28]. Moreover, the real-world data only has depth maps $\{I_j\}$, the ground-truth volumetric meshes are not available, making it hard to fine-tune the network in a supervised manner. Thus, we propose a self-supervised adaptation framework (Fig. 2) to resolve the covariate shift.

The reconstructed mesh $\hat{\mathcal{M}}$ is rendered to the image $\tilde{I}$ using a differentiable renderer, which allows gradients to propagate backward. In parallel, we use the pre-trained image VAE to reconstruct the input depth map $\hat{I}$. The network is adapted to minimize the difference between $\tilde{I}$ and $\hat{I}$.

### D. Datasets

Labeled synthetic data $\{(I_i, \mathcal{M}_i)\}$ and unlabeled real-world data $\{(I_j)\}$ are required to train the image-to-mesh projection and adapt the network among different domains.

*1) Synthetic Data:* Labeled image-mesh pairs $\{(I_i, \mathcal{M}_i)\}$ for $i \in [1, ..., N]$ can be simulated using FEM and synthetic

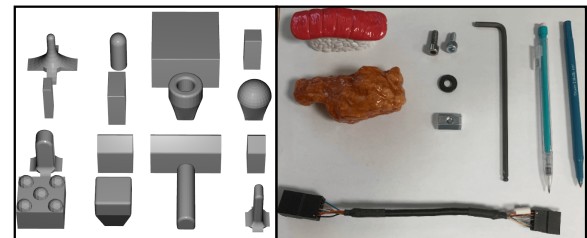

Fig. 3: *Left*: Primitive indenters. *Right*: Novel contact objects.

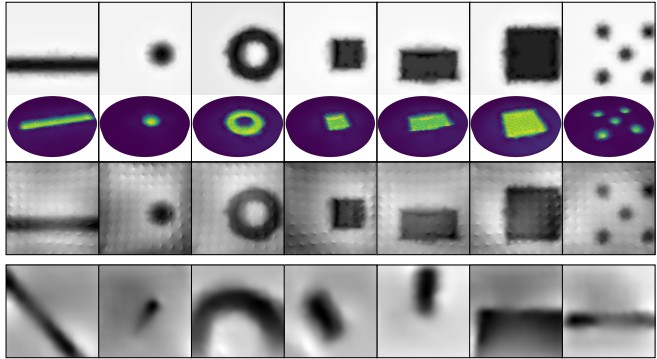

Fig. 4: Data samples. *Top*: Raw synthetic depth observations, corresponding ground-truth meshes, and augmented synthetic depth observations. *Bottom*: Real-world depth observations for sample indenters.

cameras. In this work, FEM is performed using the GPU-based Isaac Gym [29]. A FEM model for the GelSlim is created as Fig 1(b). To generate data pairs, 16 primitive indenters (Fig. 3–Left) are utilized to interact with the elastomer at randomized positions and rotations. The Isaac Gym simulator collects vertex positions $\mathcal{M}$ at each contact trajectory. The depth map $I$ is then rendered with a synthetic camera. Fig. 4 shows examples of synthetic data pairs.

*2) Real-World Data:* Real-world datasets $\{I_j\}$ are obtained with physical GelSlim sensors and various indenters (Fig. 4). Primitive indenters are 3D printed and interact with the sensor is randomized. Besides primitive shapes, several household and industrial objects are used as a novel set (Fig. 3–Right). The novel set represents common objects that the GelSlim will work with. Moreover, we use two GelSlim sensors to collect real-world data.

*3) Image Augmentations:* The appearance of synthetic images is quite different from that of real-world depth maps, as in Fig. 4. The depth reconstruction process for the physical GelSlim introduces significant noise into the image. To enhance the performance in the real world, this paper injects Perlin noise and adds a real-world reference noise image into the synthetic images [28]. The Perlin noise provides a realistic gradient for the image and imitates the real-world camera noise. The reference image provides sensor-specific noise.

In total, 1.28M unique labeled image-mesh pairs were obtained from the simulator, and 1,651 real-world images were obtained for 2 GelSlim sensors with 19 indenters.

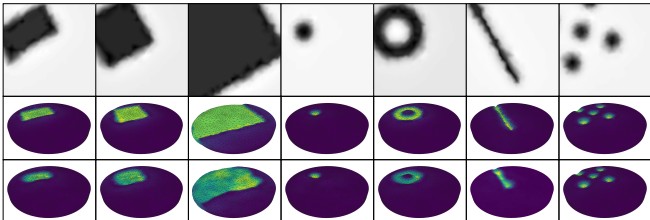

Fig. 5: Image-to-mesh projection results with synthetic data. *First row*: Input depth observations. *Second row*: Corresponding ground-truth mesh. *Third row*: Reconstructed volumetric mesh with our approach.

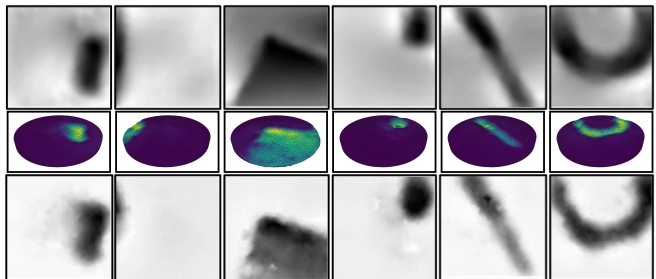

Fig. 6: Experiments with real-world primitive contact objects. *First row*: Input depth observations. *Second row*: Reconstructed volumetric meshes. *Third row*: Rendered depth images from reconstructed meshes.

## III. RESULTS

In this section, we present the experiments for supervised image-to-mesh projection and self-supervised adaptation.

### A. Supervised Projection

Our proposed supervised image-to-mesh projection was evaluated using synthetic data. The training yields $0.012cm$ root-mean-square error (RMSE) between the ground-truth vertex positions and predicted vertex positions. Fig. 5 shows a batch of projection results. The results show that the reconstruction is accurate and captures the contact information. We also investigate the usefulness of the VAE pre-training as described in II-B. We trained the image-to-mesh network from scratch and observed $0.009cm$ and $0.085cm$ training and validation errors, respectively. This suggests that the network overfits without the pre-training, which aligns with the findings presented in [30].

### B. Self-Supervised Adaptation

Section II-C and Section II-D.3 introduce a self-supervised adaptation method and synthetic data augmentations to resolve the covariate shift problem. This section shows that neither adaptation nor augmentation can achieve the objective alone. Moreover, experiments demonstrate that the proposed methods can adapt networks in different domains.

The adaptation is performed with the real-world dataset $\{I_j\}$, without ground-truth mesh availability. We use the RMSE between $\hat{I}$ and $\tilde{I}$ to evaluate the performance of the adaptations.

TABLE I: Domain adaptation results with real-world data. The root-mean-square error (RMSE) is measured between reconstructed images $\tilde{I}$ and rendered images $\hat{I}$.

| Source → Target | RMSE before/after Adaptation ($cm$) |
|---|---|
| *Sim-Prim.* → *Real-Prim* | $0.57 \rightarrow 0.12$ |
| *Sim-Prim* → *Real-Prim-2* | $0.77 \rightarrow 0.20$ |
| *Real-Prim* → *Real-Prim-2* | $0.35 \rightarrow 0.16$ |
| *Real-Prim* → *Real-Novel* | $0.64 \rightarrow 0.41$ |
| *Sim-Prim* → *Real-Novel* | $1.30 \rightarrow 0.62$ |

Networks were trained or tuned on source domains and then adapted to target domains. The RMSEs were measured before and after the adaptation.

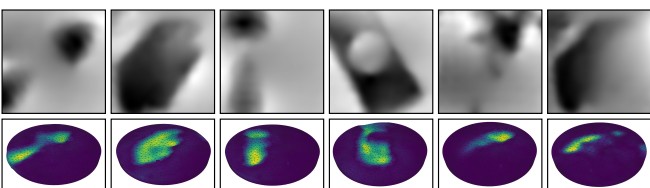

Fig. 7: Experiments with real-world novel contact objects. *First row*: Input depth observations. *Second row*: Reconstructed volumetric mesh from the network.

*1) Ablation Studies:* We compare the effects of the adaption model and the synthetic data augmentations. As a baseline, models with neither augmentations nor adaptations yield $1.03cm$ RMSE. We observe that using only adaptation ($0.79cm$ RMSE), or only augmentation ($0.57cm$ RMSE), results in lower performance. The reason for higher performance when both are used ($0.12cm$ RMSE) is two-fold. On one hand, the data augmentation enlarges the distribution of the synthetic dataset, which causes the real-world data to be within distribution (or close to). On the other hand, the adaptation model transfers the network from the simulated distribution to the real-world distribution, ensuring invariant feature encodings. A batch of qualitative reconstruction examples is shown in Fig 6.

*2) Domain Adaptations:* The network was adapted among various data domains, including simulated data with primitive contact objects (*Sim-Prim*), real-world data with primitive contact objects (*Real-Prim*), real-world data with novel contact objects (*Real-Novel*), and real-world primitive data with a second GelSlim sensor (*Real-Prim-2*).

Table I and Fig. 7 show the transfer results among domains. The results suggest that the proposed method can effectively improve the performance of the network under both visual and shape differences. However, we conclude that the covariate shifts for visual noise and shape differences are not correlated, as for experiment *Sim-Prim* → *Real-Novel* is less compared to the others. Adaptation for each separately performs better compared to adaptation for both.

## IV. DISCUSSION

This work presents a framework to synthesize volumetric meshes of vision-based tactile sensors for novel contact interactions. Our work has several contributions. First, we present a 3D FEM simulator for vision-based tactile sensors and a simulator calibration approach. Second, we generate a

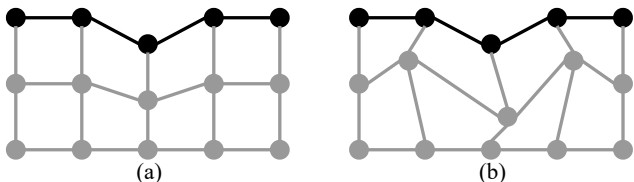

(a)                              (b)

Fig. 8: Non-injective projection of the internal vertices. Black nodes are surface vertices that the observations can supervise. Gray nodes are internal vertices that can randomly move without affecting the proposed adaptation loss. I.e., (a) and (b) will yield the same observation on the surface, while (a) is preferred.

dataset for the GelSlim sensor with both simulated and real-world contacts. Third, we propose a label-free adaptation method and image augmentations for domain transfers; this approach can effectively transfer networks to various visual and different shape scenarios. Lastly, our network efficiently reconstructs the volumetric mesh with depth images and precisely estimates the contact profiles of different shapes. More details of the method and results are available in the full version of the paper [1].

The present work also has some limitations. First, we do not constrain internal vertices during the adaptation. In the self-supervised training, the network is optimized with image observations, which only capture the surface displacement of the volumetric mesh. However, the surface displacement does not provide injective supervision for unobservable internal vertices. In other words, the internal vertices remain free-floating during the adaptation; they can move freely in the interior of the mesh without affecting the training loss, as shown in Fig 8. Such unconstraint vertices can be detrimental to reconstructing the mesh vertices using surface observation. Our experiments demonstrate that the network begins to predict random internal vertices after the first several epochs. We hypothesize that the network has some self-regularity at the beginning of the adaptation, inherited from the pre-training dataset. There is a potential solution to such a problem: adding penalty terms as a regulation. By leveraging the minimum energy principle [31], [32], it is possible to design a differentiable function that computes the energy of deformations. Such energy should be minimized simultaneously during the adaptation to mitigate the randomness of internal vertices.

The second limitation is that the current method does not predict the dynamics of the elastomer. Instead, our proposed method learns the mapping from surface observations to the mesh states. Compared to other state representations, e.g., images [13] and surface mesh [4], [5], [33], volumetric mesh contains internal vertices and edges and thus can better simulate the deformation of the objects [25], [34]. Learning the dynamics for volumetric meshes can allow model predictive control (MPC) applications and benefit reinforcement learning (RL). An MPC-based algorithm or a model-based RL agent [35] can be designed to determine actions for robotic manipulation. On the other hand, there

are many previous works on learning the dynamics for meshes [19]–[21]. These methods, however, focus on simple problem formulations in that the mesh vertices are known exactly at each timestep. A more challenging and practical scenario is to learn the dynamics with observations only since the actual mesh states are unavailable in real-world data.

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
