# OpenReview forum: "Synthesizing and Simulating Volumetric Meshes from Vision-based Tactile Imprints"
_ICRA.org/2022/Workshop/Contact-Rich — ICRA 2022 Workshop: RL for Manipulation Poster_

### Official Review · Reviewer_kHow · 2022-05-06
**Very nice paper about synthesizing volumetric meshes from vision based tactile sensors. It does not currently contain examples of RL/control but it could enable them.**

**Rating:** 7
**Confidence:** 3

**Review:**

This paper proposes a method for learning to synthesize and simulate the volumetric mesh of an object through tactile sensors. The paper is concise and very clearly presented, even in this shorter format. The experiments support the claims and it is very positive that the limitations are discussed clearly.
It would be interesting to additionally discuss computational complexity of this method, engineering effort to tune the components comprising it and training durations.

---

### Official Review · Reviewer_JRb6 · 2022-05-09
**Novel paper for learning 3D mesh from vision-based tactile sensors**

**Rating:** 7
**Confidence:** 3

**Review:**

Summary: This work proposed a framework to learn the 3D mesh from vision-based tactile sensors in both the sim and real. A VAE-based structure is applied to learn the depth-to-mesh projection while preventing overfitting. A self-supervised adaption with domain randomization is applied for sim2real. Besides, a large synthetic dataset with the FEM model is generated using Issac Gym, and a real dataset is collected with two GelSlim sensors. Experiments show the effectiveness of the proposed framework, and the approaches to bridge the sim2real gap.

Pros:
- The paper is well written with a clear problem setup, model, data, and experiments.
- It is novel to learn the 3D mesh from tactile imprint from the representation perspective.
- The work proposed to use differentiable rendering for learning the sim2real adaption and show its effectiveness.

Cons:
- As the authors mentioned in the discussion, the 3D mesh is supposed to better capture the dynamics of the elastomer. Based on the current experiments, it is not very obvious the learned mesh can benefit the dynamics learning. Could be improved in future works.

---

### Official Review · Reviewer_v7MZ · 2022-05-13
**GelSlim Volumetric Meshes - review**

**Rating:** 8
**Confidence:** 3

**Review:**

The authors present a method for generating a volumetric mesh of the elastomer on a tactile sensor using tactile imprints recorded by a GelSlim device. Their method projects an image to a mesh deformation using convolutional mesh autoencoders (COMA), and is first trained using a supervised image to mesh projection using a synthetic dataset. For dataset augmentation they add Perlin noise to the synthetic images to imitate real-world camera noise.